# CALB Immobilized on Octyl-Agarose—An Efficient Pharmaceutical Biocatalyst for Transesterification in Organic Medium

**DOI:** 10.3390/ijms26146961

**Published:** 2025-07-20

**Authors:** Joanna Siódmiak, Jacek Dulęba, Natalia Kocot, Rafał Mastalerz, Gudmundur G. Haraldsson, Tomasz Siódmiak

**Affiliations:** 1Department of Laboratory Medicine, Faculty of Pharmacy, Ludwik Rydygier Collegium Medicum in Bydgoszcz, Nicolaus Copernicus University in Toruń, 85-094 Bydgoszcz, Poland; 2Department of Pharmacy Practice, Faculty of Pharmacy, Ludwik Rydygier Collegium Medicum in Bydgoszcz, Nicolaus Copernicus University in Toruń, 85-089 Bydgoszcz, Poland; 3Department of Pharmaceutical Technology, Faculty of Pharmacy, Medical Biotechnology and Laboratory Medicine, Pomeranian Medical University in Szczecin, 71-251 Szczecin, Poland; 4Doctoral School of Medical and Health Sciences, Jagiellonian University, 31-530 Kraków, Poland; 5Department of Pharmaceutical Biochemistry, Faculty of Pharmacy, Jagiellonian University Medical College, 30-688 Kraków, Poland; 6Department of Medicinal Chemistry, Faculty of Pharmacy, Collegium Medicum in Bydgoszcz, Nicolaus Copernicus University in Toruń, 85-089 Bydgoszcz, Poland; 7Science Institute, University of Iceland, 107 Reykjavik, Iceland

**Keywords:** octyl-Sepharose CL-4B, octyl-agarose, lipase B from *Candida antarctica*, immobilization, (*R*,*S*)-1-phenylethanol, kinetic resolution, climatic chamber, thermal stability, storage stability, enantioselective transesterification

## Abstract

The growing need for developing safer and more effective methods for obtaining enantiomers of chiral compounds, particularly those with pharmacological activity, highlights the potential of biocatalysis as an appropriate pharmaceutical research direction. However, low catalytic activity and stability of free enzymes are often among the substantial limitations to the wide application of biocatalysis. Therefore, to overcome these obstacles, new technological procedures are being designed. In this study, we present optimized protocols for the immobilization of *Candida antarctica* lipase B (CALB) on an octyl- agarose support, ensuring high enantioselectivity in an organic reaction medium. The immobilization procedures (with drying step), including buffers with different pH values and concentrations, as well as the study of the influence of temperature and immobilization time, were presented. It was found that the optimal conditions were provided by citrate buffer with a pH of 4 and a concentration of 300 mM. The immobilized CALB on the octyl-agarose support exhibited high catalytic activity in the kinetic resolution of (*R*,*S*)-1-phenylethanol via enantioselective transesterification with isopropenyl acetate in 1,2-dichloropropane (DCP), as a model reaction for lipase activity monitoring on an analytical scale. HPLC analysis demonstrated that the (*R*)-1-phenylethyl acetate was obtained in an enantiomeric excess of ee_p_ > 99% at a conversion of approximately 40%, and the enantiomeric ratio was E > 200. Thermal and storage stability studies performed on the immobilized CALB octyl-agarose support confirmed its excellent stability. After 7 days of thermal stability testing at 65 °C in a climatic chamber, the (*R*)-1-phenylethyl acetate was characterized by enantiomeric excess of ee_p_ > 99% at a conversion of around 40% (similar values of catalytic parameters to those achieved using a non-stored lipase). The documented high catalytic activity and stability of the developed CALB-octyl-agarose support allow us to consider it as a useful tool for enantioselective transesterification in organic medium.

## 1. Introduction

In recent years, the importance of chiral drugs in pharmaceutical and biomedical sciences has increased significantly. Scientific progress has enabled the discovery of many active substances as racemates and their enantiomers. Additionally, the differences in pharmacological activity and side effects of the two optical isomers of a given drug molecule have been well documented [1,2]. Therefore, the development of novel methods for obtaining and analyzing optically pure compounds has become an important area of pharmaceutical research [3,4]. It should be noted that the emphasis is on methods that are characterized by a greener approach to the natural environment and do not require hazardous compounds [5]. Contemporary, enzyme-based strategies are the solutions to these concerns, mainly due to their wide range of applications, among others, synthetic chemistry as biocatalytic alternatives to existing synthetic procedures or, in the chiral resolution of racemic substrates [6]. The most commonly utilized enzymes in biocatalysis are lipases. Most of them contain a mobile lid domain, located over the active site. Thanks to this (the lid domain), lipase can change its conformation from closed (inactive) to open (active) state of the enzyme, at an aqueous–oil interface in the solution. This results in an increased exposure of the active site, enabling access of solvent and substrates. This phenomenon is called ‘interfacial activation’ [7,8]. It should be emphasized that lipases exhibit stability in organic solvents and can perform without cofactors. Moreover, due to their chemoselectivity, regioselectivity, and enantioselectivity, lipases are selected as valuable biocatalysts in catalyzing numerous reactions of pharmaceutical importance [9,10].

The lipase B from *Candida antarctica* (CALB) is one of the most applied biocatalysts for obtaining enantiomerically pure compounds [11,12]. The presence of the lid remains a controversial topic and is constantly a matter of scientific research. Wang et al. [13] stated that CALB has one small lid, and the phenomenon known as interfacial activation can be observed. In addition, it has been described in the literature that CALB in the open state has a “lid scaffold” structure, creating a self-activation mechanism [13,14]. The application of CALB as a biocatalyst in various chemical reactions is broadly discussed in the literature. Chen et al. [15] used CALB to catalyze the regioselective acetylation of phloridzin, a compound with, among others, antioxidant, anti-inflammatory, and antitumor activities. Pérez-Venegas et al. [16] described a mechanoenzymatic kinetic resolution procedure with the utilization of CALB to achieve enantiomers of (*R*,*S*)-ketorolac in high enantiopurity. Zappaterra et al. [17] performed the enzymatic esterification of the ursodeoxycholic acid (UDCA) with glycerol, applying an immobilized CALB (Novozym-435) in solvent-free and solvent-assisted systems. UDCA is used in the treatment of primary biliary cirrhosis and dissolving cholesterol gallstones [17,18]. Another compound, with diverse applications, among others, in the pharmaceutical (as a chiral building block) and cosmetic industry (as a fragrance additive) is (*R*,*S*)-1-phenylethanol [19,20,21,22]. This compound can be resolved into single enantiomers in kinetic resolution by transesterification catalyzed by CALB (Figure 1). It is worth mentioning that the kinetic resolution is characterized as highly enantioselective when the difference in the enantiomer reaction rates, enantiomeric ratio E (k_R_/k_S_), is above or equal to 100 (E ≥ 100) [23]. In the literature, examples of using CALB as a biocatalyst in the kinetic resolution of (*R*,*S*)-1-phenylethanol can be found [24,25].

In the scientific literature, there is a constant growth in the number of proposed protocols enabling lipase catalytic activity and stability to be increased. One of the strategies that can beneficially influence the assessed catalytic parameters is immobilization of the biocatalyst. This process, among others, may simplify enzyme recovery and enhance control of the biocatalytic reaction [26]. As mentioned earlier, CALB is recognized to have a phenomenon of interfacial activation. Therefore, the immobilization by interfacial activation on hydrophobic supports may be a method of choice. This technique is a simple, direct, low-cost, and highly efficient strategy, enabling purifying, immobilizing, stabilizing, and hyperactivating lipases [27]. Importantly, lipases tend to adsorb onto various hydrophobic surfaces, such as hydrophobic support, hydrophobic protein, or the open form of other lipases [28].

Octyl-agarose beads are one of the supports that allow the biocatalyst immobilization, based upon the lipase mechanism of action, i.e., interfacial activation. It has been shown that lipases immobilized via interfacial activation on hydrophobic supports are typically more stable. However, there are also some drawbacks associated with their use, e.g., in the presence of detergents or high concentrations of hydrophobic organic cosolvents, the enzyme protein may be released from the support. It is noteworthy that the octyl-agarose beads can serve as a very useful support, among others, as they can undergo various modifications, enabling the production of heterofunctional supports [29]. An example of a commercially available support made up of octyl groups coupled to a cross-linked 4% agarose matrix is the Octyl-Sepharose CL-4B support. The beneficial effect of immobilized CALB onto Octyl-Sepharose CL-4B on its lipolytic activity [30,31] and enantioselectivity [32] has been presented in our previous papers. A simplified mechanism of lipase immobilization by interfacial activation is shown in Figure 2.

In view of the relatively low number of studies dealing with the usefulness of the CALB-octyl-agarose support applied in reactions in an organic solvent, we decided to conduct further research in this area. Therefore, the presented results contribute significantly to the discussion on the utility of the tested immobilized biocatalyst in organic reaction medium.

In this work, the transesterification in an organic solvent (1,2-dichloropropane, DCP) of (*R*,*S*)-1-phenylethanol catalyzed by CALB immobilized onto octyl-agarose support to achieve the (*R*)-1-phenylethyl acetate has been performed. Based on the results of enantiomeric excess, enantiomeric ratio, and conversion values, the enantioselectivity of immobilized CALB was evaluated. The optimization of the immobilization of CALB using a range of buffers with varied pH values and concentrations, as well as the study of the influence of immobilization time and temperature, was carried out. Moreover, the thermal and storage stability tests of the supports with immobilized CALB in a climatic chamber and refrigerator were conducted.

## 2. Results and Discussion

### 2.1. Optimization of Candida Antarctica Lipase B (CALB) Immobilization

#### 2.1.1. Influence of Buffer pH

In this study, the octyl-agarose support for lipase immobilization via interfacial activation was used. The optimization of immobilization was carried out in buffers (100 mM) with different pH values: citrate (pH 4, pH 5, and pH 6), phosphate (pH 7 and pH 8), and Tris base (pH 9). Both for immobilization and transesterification, polypropylene vials were applied. The transesterification was performed in 1,2-dichloropropane (DCP) as a reaction medium. The results of lipase activity expressed as the enantiomeric excess of the substrate (ee_s_), enantiomeric excess of the product (ee_p_), conversion (C), and enantiomeric ratio (E) are presented in Table 1 and Figure 1. During the preliminary phase of this study, we investigated the effect of various organic solvents as a reaction medium (*n*-hexane, *n*-heptane, dichloromethane (DCM), 1,2-dichloroethane (DCE), 1,2-dichloropropane (DCP), diisopropyl ether (DIPE), and *t*-butyl methyl ether (MTBE)) on the assessed catalytic parameters of immobilized CALB. Among the tested solvents, the best results were obtained for DCP; hence, this solvent was used in further studies.

The data gathered in Table 1 and Figure 1 indicate that with a decrease in the pH value of the buffer used for lipase immobilization, there is an increase in the value of conversion. Utilizing citrate buffer (pH 4) during the immobilization procedure allowed for the achievement of the highest conversion values (C = 40.8 ± 0.8%). After 24 h of kinetic resolution of (*R*,*S*)-1-phenylethanol by enantioselective transesterification with isopropenyl acetate as an acylating agent, the (*R*)-1-phenylethyl acetate was obtained in an enantiomeric excess of ee_p_ > 99%. The enantioselectivity of the transesterification was very high (E > 200). The lipase immobilization in the buffer at pH 4 offered more than a 2-fold increase in conversion compared with the buffer at pH 9. The immobilization yield (*Iy*) was calculated: in the case with the use procedure immobilization with citrate buffer (pH 4; 100 mM) the immobilization yield was *Iy* = 32.4 ± 0.4%, using the phosphate buffer (pH 7; 100 mM) *Iy* = 21.1 ± 0.7%, and with the Tris base buffer (pH 9; 100 mM) *Iy* = 15.3 ± 0.3%.

It can be concluded that the pH value and nature (chemical composition) of the buffer are essential factors in gaining optimal immobilization conditions for the tested lipase (optimal structural conformation). Immobilization of CALB on octyl-agarose beads in citrate buffer (pH 4) may have a stabilizing impact on the active site of the biocatalyst. It is assumed that lipase catalytic activity can be altered by the buffer influence on the net charge of the enzyme, the ionization state of the catalytic triad, as well as on the interaction between CALB and the support [33,34]. The loading of CALB on octyl-agarose support may be another significant factor affecting the lipase activity. As was presented in our previous paper [32], the loaded amount of immobilized lipase was higher under acidic pH buffered conditions than when basic conditions were used. We believe that a higher amount of immobilized lipase may probably contribute to the higher conversion values achieved in the transesterification in the organic solvent. It was pointed out, based on the results of inactivation tests (performed in different buffers), that the nature of the buffer impacts the stability of lipase immobilized via interfacial activation [35]. A separate study indicated that the enzyme loading (probably by intermolecular interaction) and the buffer nature may influence the enzyme stability, specificity, or enzyme catalytic activity [36].

Based on the data presented in this study and our prior papers [31,32], the relationship between lipase catalytic activity and the reaction medium (aqueous or organic solvent), substrates, and applied immobilization buffers (pH values) can be observed. As was mentioned above, high conversion values of enantioselective transesterification of (*R*,*S*)-1-phenylethanol with isopropenyl acetate in 1,2-dichloropropane were obtained when citrate buffer (pH 4) was used for CALB immobilization. Similarly, as previously described [31], the highest activity (lipolytic) values for CALB, expressed as activity recovery, were reached when citrate buffer at pH 4 for immobilization was applied. On the other hand, as we presented in [32], the Tris base buffer (pH 9) was noted to offer the highest results for the esterification of (*R*,*S*)-flurbiprofen with methanol. Thus, the results discussed above suggest that the crucial elements for designing the CALB immobilization conditions are the type of reaction (including the medium) in which lipase will be applied and the chemical structure of substrates. 

Considering the obtained conversion values, enantiomeric excess, as well as enantiomeric ratio, it was decided that the immobilization protocol with citrate buffer would be employed in the following phase of the project.

#### 2.1.2. Influence of Buffer Concentration, Immobilization Time, and Temperature

The immobilization of CALB was performed in a citrate buffer (pH 4) of different concentrations (50 mM, 100 mM, 300 mM, and 500 mM). The reaction vials were made of polypropylene material. The transesterification was conducted in 1,2-dichloropropane (DCP) as the medium. The results are presented in Table 2 and, like in Table 1, are expressed as the enantiomeric excess of substrate (ee_s_), enantiomeric excess of product (ee_p_), conversion (C), and enantiomeric ratio (E).

According to the data presented in Table 2, there was no significant effect noticed for the buffer concentration used for immobilization on the enantioselectivity of CALB in the kinetic resolution of (*R*,*S*)-1-phenylethanol with isopropenyl acetate. High values for the evaluated catalytic parameters were consistently achieved in all tested concentrations (50 mM, 100 mM, 300 mM, and 500 mM). Assumingly, such high values for enzyme enantioselectivity achieved in 1,2-dichloropropane under the conditions of the tested buffer concentrations (used for immobilization) result from the favorable orientation of the enzyme during immobilization. It is noteworthy that the results described in our previous publication [32], where the enantioselective esterification of (*R*,*S*)-flurbiprofen with methanol was conducted, indicated a substantial effect of the concentration of the buffer (Tris base) applied for immobilization. The presented differences in the impact of buffer concentration on lipase activity confirm our observation that there are no universal CALB immobilization conditions on the tested support. As mentioned earlier, when optimizing the immobilization conditions, consideration should be given to the reaction type, the reaction medium, and the substrate molecules.

In our previous papers [31,32], we broadly discussed the effect of the buffer applied for immobilization on the CALB activity in reactions performed in both aqueous and organic media with different substrates (olive oil and flurbiprofen). It was observed that the concentration of the buffer used for immobilization influenced the CALB activity. The aforementioned effect may vary depending, among others, on the substrates used in the reaction. It should be noted that in the solution, ions are a crucial factor that affects, inter alia, the solubility, stability, and surface charge of proteins, as well as viscosity, and the tension of the surface of the aqueous medium. In the buffer itself, some ions may influence the substrate-to-protein or protein-to-protein interactions (electrostatic). Additionally, it should be highlighted that ionic-strength-dependent molecular forces are intrinsically controlled by dissociation and hydration processes [37]. In the current paper, the data have been discussed in terms of the pH values and buffer concentrations and the chemical composition (nature) of buffers applied in the immobilization procedure; however, it should be noted that the ionic strength of the buffer used for the immobilization may also affect the enzyme activity.

As part of the preliminary studies in our laboratory, the immobilization time was also optimized. Samples in a citrate buffer (pH 4; 300 mM) were mixed for 5 min at room temperature and then kept at 4 °C for 2, 8, 10, 14, and 20 h. The results of the conducted model reaction (transesterification of (*R*,*S*)-1-phenylethanol) indicated that the optimal immobilization time was 14 h, as this condition ensured the highest conversion values. Extending the incubation time beyond 14 h did not lead to further increases in conversion and enantioselectivity (after 2 h of immobilization C = 14.3 ± 0.2%; after 8 h C = 30.3 ± 1.1%; after 10 h C = 38.4 ± 1.3%, after 14 h C = 43.1 ± 0.5%, and after 20 h C = 43.6 ± 1.9%). Another parameter optimized was the immobilization temperature. Samples (in citrate buffer at pH 4; 300 mM) were mixed for 5 min at room temperature, followed by incubation (CALB) for 14 h at 4 °C, 22 °C, and 37 °C. The analyses demonstrated that the highest conversion values of kinetic resolution were achieved when immobilization was performed in a refrigerator (4 °C). Conversion values at the other temperatures were slightly lower than those gained from lipase immobilized in a refrigerator (C = 41.1 ± 1.0% at 22 °C and C = 40.4 ± 0.8% at 37 °C).

We also reviewed recent publications on lipase immobilization on agarose support (Table 3), identifying the lipases used, supports (with their modifications), substrates for activity assessment, reaction medium, and type of reaction. It should be noted that the literature emphasizes the fact that the application of lipases immobilized on agarose in organic solvents is poorly described [38]. As can be seen from the prepared table, most studies on the catalytic activity of immobilized lipases (on the discussed support—agarose), particularly CALB, have been performed in an aqueous medium (mainly hydrolysis reactions in buffer). In contrast, our study proposes the use of octyl-agarose-immobilized CALB for reactions in organic solvents. This approach opens new possibilities for conducting biocatalysis by esterification or transesterification, using CALB immobilized on octyl-agarose, offering excellent enantioselectivity. Our protocol for the preparation of octyl-agarose support with CALB for biocatalysis in an organic reaction medium, including immobilization of CALB on octyl-agarose in acidic pH buffer—citrate buffer (pH 4; 300 mM)—followed by air-drying, according to our knowledge, has not been widely described in the literature. The immobilized CALB obtained utilizing this method can be introduced into organic media and used for kinetic resolution of (*R*,*S*)-1-phenylethanol via transesterification (in DCP; logP value 1.98), with high enantioselectivity. At the same time, we emphasize the potential to modulate the CALB activity by adjusting the pH of the immobilization buffer. Therefore, immobilization conditions (e.g., including buffer pH and concentration) should be selected in relation to the substrate, type of reaction, and organic solvent applied in biocatalysis.

Considering the results, it can be stated that the CALB-octyl-agarose support can potentially be treated as a useful tool in pharmaceutical biocatalysis to perform transesterification in an organic solvent.

### 2.2. Thermal Stability, Storage Stability, and Operational Stability Study of CALB

To evaluate the thermal stability of the catalytic protein, a thermal stability study of the CALB in its dry form was performed in a climatic chamber. These studies were carried out for 7 days. In the climatic chamber, the influence of a temperature of 65 °C was determined, and additionally, under this study, the effect of visible light (wavelength range: 400–800 nm) was also examined. For comparison with extremely high temperatures, a parallel experiment was performed at 4 °C in a refrigerator, without exposure to visible light. This setting was used as a control to assess the impact of high temperature (65 °C) and light (400–800 nm) on the thermal stability of the immobilized enzyme. After 7 days, the immobilized biocatalysts were used for the kinetic resolution of (*R*,*S*)-1-phenylethanol with isopropenyl acetate by transesterification (reaction time was 24 h) in DCP. The catalytic data were compared with the results obtained for biocatalysts that were not stored (applied in the kinetic resolution after immobilization and drying). The immobilization was performed in a citrate buffer (pH 4; 300 mM). Polypropylene reactors were used both for immobilization as well as kinetic resolution. The results are presented in Table 4.

From analyzing the results obtained for the catalytic parameters (see Table 4), it is evident that the created immobilized biocatalysts exhibit excellent thermal stability, as the data on lipase enantioselectivity upon storage are comparable to those preparations that were not stored in a climatic chamber or refrigerator. There was no observed negative impact from the storage (in a climatic chamber and a refrigerator) of the immobilized CALB (in dry form) on the catalytic activity expressed as conversion, enantiomeric excess, and enantiomeric ratio. We noticed that the immobilized lipase, after storage in a climatic chamber at extremely high temperature (65 °C) for 7 days in closed polypropylene vials (both protected and exposed to the light) presented similar values for the catalytic activity as the immobilized CALB not stored in the climatic chamber (C = 42.5 ± 0.8%, ee_p_ = 99.9 ± 0.1%; C = 42.9 ± 0.4%, ee_p_ = 99.7 ± 0.3%; and C = 43.1 ± 0.5%; ee_p_ = 99.8 ± 0.2%, respectively) (Figure 2). Likewise, the biocatalysts stored in the refrigerator (in closed vials), protected from light, exhibited similar catalytic activity, compared with those that had not been stored (C = 39.4 ± 0.8%, ee_p_ = 99.7 ± 0.2%; and C = 43.1 ± 0.5%, ee_p_ = 99.8 ± 0.2%, respectively). These findings suggest a potentially favorable effect of the immobilization conditions for the CALB (citrate buffer (pH 4; 300 mM) in polypropylene vials) on both CALB thermal stability and catalytic activity in the kinetic resolution. We believe that the immobilized CALB stored in a climatic chamber or refrigerator can maintain its open form.

It should be noted that the results obtained in this study demonstrate a similar trend to those presented in our previous papers [30,31,32], in which we pointed out that, after storage, apart from good stability of the CALB immobilized on octyl-agarose, often an increase in catalytic activity occurs. The data in this paper do not show an increase in the CALB catalytic activity after storage. However, the findings exhibit excellent thermal stability of the immobilized biocatalyst. There is a limited number of studies that examine the stability of the CALB immobilized onto octyl-agarose, based on catalytic parameters of the kinetic resolution conducted in organic solvents. Therefore, the presented results contribute to the discussion on unified methods for testing the thermal stability of catalytic systems. As was previously mentioned [30,31,32], we assume that good stability may result from the elements of the CALB structure, including the lid. Additionally, it is also suggested that the stability under the tested conditions can be related, among others, to the enzyme’s conformation.

In the next step, a storage stability study of CALB was conducted. The immobilized CALB, using the same immobilization procedure as in the thermal stability study, was stored in dry form in two conditions: at 4 °C in a refrigerator (without light in the visible spectral range) and at room temperature (22 °C, without light in the visible spectral range) for 28 days. Storage stability was monitored at four time points: after 7, 14, 21, and 28 days of storage. The assessment of storage stability was carried out using the same reaction conditions as in the thermal stability study. The results of the studies (Figure 3) indicate good stability of the catalytic system under both storage conditions throughout the 28-day period, stored both at 4 °C and 22 °C.

Summarizing, the results from the stability tests point out that through the application of optimal immobilization protocols, we have managed a successful development of catalytic systems with excellent thermal and storage stability.

As part of the project, operational stability studies were also conducted by reusing the biocatalyst in subsequent catalytic cycles of the transesterification of (*R*,*S*)-1-phenylethanol. After the third cycle, the conversion of the reaction was C = 10 ± 0.3%, which indicates a significant decrease in conversions compared to the initial value (C = 43.1 ± 0.5% from the first cycle). We believe that the obtained results were influenced by technological difficulties with support recovery from reaction medium, the analytical scale of the experiments (50 mg of support), and the nature of the interactions between CALB and the support (suggesting possible enzyme leakage). Immobilization of lipase was performed in a citrate buffer (pH 4; 300 mM).

## 3. Materials and Methods

### 3.1. Materials

Octyl-Sepharose CL-4B (GE Healthcare, Uppsala, Sweden), (*R*,*S*)-1-phenylethanol, (*R*)-1-phenylethanol, (*S*)-1-phenylethanol, *n*-heptane, *n*-hexane, diisopropyl ether (DIPE), *t*-butyl methyl ether (MTBE), 2-propanol, trifluoroacetic acid, 1,2-dichloropropane (DCP), isopropenyl acetate, hydrochloric acid, and Tris base reagent were gained from Sigma-Aldrich (Steinheim, Germany). Dichloromethane (DCM), 1,2-dichloroethane (DCE), citric acid monohydrate, disodium hydrogen phosphate dihydrate, sodium dihydrogen phosphate monohydrate, and molecular sieve 4 Å were purchased from POCH (Gliwice, Poland). Trisodium citrate was from Chempur (Piekary Śląskie, Poland). The lipase B from *Candida antarctica* (CALB, produced in yeast was from ChiralVision—Leiden, The Netherlands).

### 3.2. Instrumentation and Chromatographic Conditions

The water used in this investigation was purified by the Milli-Q Water Purification System (Millipore, Bedford, MA, USA). The thermal stability study of the lipase in the immobilized form was performed in a climatic chamber KBF P240 (Tuttlingen, Germany). The buffers were prepared by use of a SevenMulti pH meter (Mettler-Toledo, Schwerzenbach, Switzerland). The Octyl-Sepharose CL-4B support was prepared by an Eppendorf MiniSpin Plus centrifuge (Hamburg, Germany) and a vortex Velp Scientifica ZX4 mixer (Usmate, Italy). The incubation of the samples was carried out in a Thermomixer comfort (Eppendorf AG, Hamburg, Germany). The analysis of the kinetic resolution of (*R*,*S*)-1-phenylethanol was performed with the application of HPLC. The Shimadzu HPLC system (Kyoto, Japan) was composed of a pump (LC-20 AD), a UV–VIS detector (SPD-20A), a degasser (DGU-20A_5R_), an autosampler (SIL-20AC_HT_), and a column oven (CTO-10AS_VP_). As a chiral separator, a Lux Cellulose-3 (LC-3) (4.6 mm × 250 mm) column with cellulose tris(4-methylbenzoate) and pre-column (Guard Cartridge System, KJO-4282 model) was used. The column had a 5 μm particle size. The optimal chromatographic conditions for (*R*)- and (*S*)-1-phenylethanol and their esters were established with an *n*-heptane/2-propanol/trifluoroacetic acid (98.7/1.3/0.15, v/v/v) mobile phase at a flow rate of 1 mL/min. The UV detection wavelength was set at 254 nm. The temperature of the chromatographic process was 15 °C.

The enantiomeric excesses of the substrate (ee_s_) and the product (ee_p_), the enantiomeric ratio (E), as well as the conversion (C), were calculated using the following equations [47,48].

The ee_s_ and ee_p_ values were expressed as follows:$$\left(\%\right){ee}_{s}=\frac{\left|R_{s}-S_{s}\right|}{R_{s}+S_{s}}\times 100$$$$\left(\%\right){ee}_{p}=\frac{\left|R_{p}-S_{p}\right|}{R_{p}+S_{p}}\times 100$$

R_s_, S_s_—enantiomers of the substrate ((*R*,*S*)-1-phenylethanol) represent the peak areas of the *R*- and *S*-enantiomers, respectively.

R_p_, S_p_—enantiomers of the product ((*R*,*S*)-1-phenylethyl acetate) represent the peak areas of the *R*- and *S*-enantiomers, respectively.

The conversion (C):$$\left(\%\right)C=\frac{{ee}_{s}}{{ee}_{p}+{ee}_{s}}\times 100$$

The enantiomeric ratio (E):$$E=\frac{\ln\left[\left(1-C\right)\left(1-{ee}_{s}\right)\right]}{ln[\left(1-C\right)\left(1+{ee}_{s}\right)]}$$

### 3.3. Methods

#### 3.3.1. Preparation of the Octyl-Agarose Support

The method of support preparation was taken from manufacturer data and papers [30,31,32]. The octyl-agarose beads suspension (110 μL) was transferred to a polypropylene tube. In the next step, 1 mL of filtered water was inserted into the tube with the support suspension, and the mixture was stirred by applying a vortex for 3 min, followed by centrifugation for 15 min at 9000 rpm. Subsequently, the support, after separation from the supernatant, was weighed (50 mg).

#### 3.3.2. Immobilization of CALB onto Octyl-Agarose Support

The immobilization method was performed in our laboratory with minor changes [30,31,32]. The simplified procedure is presented in Figure 3. An amount of 10.0 mg of the CALB was placed in an Eppendorf tube (2.0 mL) and suspended in 1.0 mL of an appropriate buffer. The sample was allowed to stay for 15 min at room temperature. After this time, the CALB suspension was mixed and then placed into the polypropylene vial (2.0 mL) containing 50 mg of the prepared octyl-agarose support. The samples were mixed for 5 min and then maintained at 4 °C for 14 h. Finally, the supernatant was removed, and the supports with the immobilized CALB were air-dried for 48 h. The procedures were performed in triplicate. For immobilization, the following buffers were used: citric buffer—pH 4 (50 mM, 100 mM, 300 mM, and 500 mM), pH 5 (100 mM), pH 6 (100 mM); phosphate buffer—pH 7 (100 mM), pH 8 (100 mM); and Tris base buffer pH 9 (100 mM). The immobilization time was tested. Samples in a citrate buffer (pH 4; 300 mM) were mixed for 5 min at room temperature and then maintained at 4 °C for 2, 8, 10, 14, and 20 h. Temperature optimization was also performed. After mixing samples (in a citrate buffer—pH 4; 300 mM) for 5 min at room temperature, CALB was incubated for 14 h at three different temperatures: 4 °C, 22 °C, and 37 °C.

The immobilization yield (*I_y_*) was calculated by the following equation:$$I_{y}=\frac{{LA}_{B}}{{LA}_{10}}\times 100\%$$
where *I_y_* is the immobilization yield, *LA*_B_ is the amount of lipase being the difference between the starting amount of CALB and the amount remaining in the supernatant after immobilization onto 50 mg of support (data from Table 1, lipase loading), and *LA*_10_ is the starting amount of lipase (10 mg) [20].

#### 3.3.3. Kinetic Resolution of (*R*,*S*)-1-Phenylethanol Catalyzed by CALB

The kinetic resolution of (*R*,*S*)-1-phenylethanol was performed according to the method described in the literature [19,20,49], including some changes. The simplified procedure is presented in Figure 3. The (*R*,*S*)-1-phenylethanol (0.08 mM), reaction medium (DCP; 410 μL), isopropenyl acetate (0.3 mM), and molecular sieve 4 Å were added to the polypropylene vial containing the octyl-agarose support with the immobilized lipase (CALB). The sample vials were closed and secured with thermal insulation tape and then incubated in a Thermomixer with mixing (37 °C, 550 rpm). The reaction time was 24 h. Afterwards, the samples (50 μL) were collected and evaporated at room temperature, dissolved (*n*-heptane, 0.9 mL), and after filtration (0.45 μm), were injected (5 μL) on the HPLC column. Analyses were performed in triplicate. At the stage of preliminary studies, organic solvents were used as a reaction medium: *n*-hexane, *n*-heptane, DCM, DCE, DCP, DIPE, and MTBE.

#### 3.3.4. Stability Tests of CALB

The thermal stability study of the CALB was performed according to the literature method [30,31,32]. After immobilization in the citrate buffer (pH 4; 300 mM), the supernatant was collected, and the supports with the immobilized lipase were air-dried for 48 h. Then, the octyl-agarose beads with the immobilized enzyme were stored in polypropylene vials in a KBF P240 climatic chamber or refrigerator (4 °C). In the climatic chamber, the temperature throughout this study was maintained at 65 °C, and the visible spectral range was 400–800 nm. The immobilized lipase was stored for 7 days. Then, the enantioselectivity of the immobilized lipase was evaluated according to Section 3.3.3. The reaction time was 24 h.

a.Without storage—used after immobilization, without a storage procedure;b.Climatic chamber (65 °C, no light)—storage temperature of 65 °C, without light in the visible spectral range;c.Climatic chamber (65 °C, Vis 400–800 nm)—storage temperature of 65 °C, light in the visible spectral range;d.Refrigerator (4 °C, no light)—storage temperature of 4 °C, without light in the visible spectral range.

In the storage stability study, the immobilized CALB, using the same immobilization procedure as in the thermal stability study, was stored in dry form at 4 °C in a refrigerator (without light in the visible spectral range) and at room temperature (22 °C, without light in the visible spectral range) for 28 days. Stability was monitored at four time points: after 7, 14, 21, and 28 days of storage. The assessment of storage stability was carried out using a model transesterification reaction of (*R*,*S*)-1-phenylethanol (as described in Section 3.3.3). Each reaction was carried out for 24 h.

Operational stability studies were conducted by reusing the biocatalyst in subsequent catalytic cycles of the transesterification of (*R*,*S*)-1-phenylethanol (one reaction cycle was 24 h). Immobilization conditions and transesterification reaction conditions were the same as in the thermal and storage stability study.

## 4. Conclusions

The study of the CALB-octyl-agarose support to determine its usability as a potential tool in transesterification was performed by the kinetic resolution of (*R*,*S*)-1-phenylethanol with isopropenyl acetate as an acylating agent in DCP as a reaction medium. The reactions exhibited high enantioselectivity (E > 200), with the (*R*)-1-phenylethyl acetate achieving an enantiomeric excess (ee_p_) greater than 99% at approximately 40% conversion. This paper presents procedures for CALB immobilization on octyl-agarose support, the study of created immobilized biocatalysts catalytic activity in enantioselective transesterification, and also the study of its thermal stability (in a climatic chamber), storage stability, and operational stability. It was demonstrated that the pH and the concentration of the buffer used for immobilization of CALB on octyl-agarose support may be crucial factors influencing the enantioselectivity of the achieved immobilized form of the biocatalyst. Application of the citrate buffer at pH 4 and a concentration of 300 mM allowed us to gain optimal immobilization conditions for reaching high lipase enantioselectivity. The thermal and storage stability tests performed exhibited excellent stability of the immobilized CALB. After 7 days of storage at 65 °C, the (*R*)-1-phenylethyl acetate was characterized by an enantiomeric excess ee_p_ > 99% at a conversion of approximately 40% (values comparable to those attained using a non-stored lipase). The documented high catalytic activity and stability of the developed immobilized CALB allow us to consider the CALB-octyl-agarose support as an extremely useful tool for enantioselective transesterification in organic medium.

## Data Availability

The data are contained within this article.

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
