# Peer review of "CALB Immobilized on Octyl-Agarose—An Efficient Pharmaceutical Biocatalyst for Transesterification in Organic Medium"

_ijms, 2025, doi:10.3390/ijms26146961_

Round 1

Reviewer 1 Report

Comments and Suggestions for Authors

In this study, the authors described the newly developed protocol of immobilization of lipase B from Candida antarctica on an octyl-agarose support, the key founding of the MS, than its use in the acetylation of racemic 1-phenylethanol with isopropenyl acetate in 1,2-dichloropropane (reaction medium), somehow fragmentary work. Although the promising results (enantioselectivity E >200, eeP >99% at conv< 43%), the progress of the reaction was not analyzed after 24 h (conv< 43%), thus no further data with regard to the eeS, eeP, conv, E after 24h reaction time. A preparative-scale reaction, under the optimal reaction conditions should have also been performed and described. Thus, there is no isolation of the product(s), nor adequate characterization of its (them). 

All together, the results of this paper are worth to be published in IJMS, but only after major revision.

Suggestions:

  1. A preparative-scale kinetic resolution of racemic 1-phenylethanol through acetylation with isopropenyl acetate in 1,2-dichloropropane (reaction medium) in the presence of CALB immobilized on an octyl-agarose support, under the optimal reaction conditions should be performed and described. Adequate characterization of the isolated products is necessary (exact mass, yield, ee, optical rotation, 1H- and 13C-NMR, HRMS or EA, etc). When a product known in literature, the authors should give the literature data comparatively in the Experimental.
  2. HPLC chromatograms with baseline separation of the enantiomeric peaks for racemic alcohol and ester should be added (Supporting information). Add also the chromatograms for the enantiomeric products, from the enzymatic reaction (Supporting information).
  3. In the main text (line 338) ”The most optimal chromatographic conditions…“ should be corrected into ”The optimal chromatographic conditions…“; in Tables 1-3, columns Time of kinetic resolution should be deleted and 24 h reaction time included in the Titles). Typos, spelling errors should be corrected too.

Author Response

Authors’ Response to Reviewer 1

Comments and Suggestions for Authors

In this study, the authors described the newly developed protocol of immobilization of lipase B from Candida antarctica on an octyl-agarose support, the key founding of the MS, than its use in the acetylation of racemic 1-phenylethanol with isopropenyl acetate in 1,2-dichloropropane (reaction medium), somehow fragmentary work.

Response to Reviewer Comments: Thank you for your comment. The primary objective of this project is to investigate, on an analytical scale, the enantioselective activity of Candida antarctica lipase B (CALB) immobilized on octyl-agarose, applied in transesterification reactions carried out in an organic medium. While the majority of studies on the catalytic performance of immobilized lipases—particularly CALB immobilized on agarose supports—have focused on aqueous systems (primarily hydrolysis reactions in buffered media), our work presents a different approach. Specifically, we propose the application of octyl-agarose-immobilized CALB in organic solvents. This strategy broadens the potential of biocatalysis, enabling efficient esterification and transesterification reactions with high enantioselectivity, thereby extending the applicability of immobilized (on agarose) enzymes in organic solvents.

Although the promising results (enantioselectivity E >200, eeP >99% at conv< 43%), the progress of the reaction was not analyzed after 24 h (conv< 43%), thus no further data with regard to the eeS, eePconv, E after 24h reaction time. A preparative-scale reaction, under the optimal reaction conditions should have also been performed and described. Thus, there is no isolation of the product(s), nor adequate characterization of its (them). 

Response to Reviewer Comments: Thank you for your valuable suggestions. The kinetic resolution of racemic 1-phenylethanol, which was employed in this study as a model reaction to evaluate enantioselectivity, theoretically yields a maximum conversion of 50% (in contrast to dynamic kinetic resolution, which may achieve up to 100% conversion). Based on preliminary investigations and to standardize the presentation of results, we established a fixed reaction time of 24 hours (at this point, conversion exceeded 40%). It is important to emphasize that the objective of the study was not the preparative-scale synthesis of enantiomers of 1-phenylethanol, but rather the assessment of the enantioselective activity of the developed biocatalytic system under analytical-scale conditions. Extending the reaction time beyond 24 hours in a classical kinetic resolution setup would result in conversion levels exceeding 50%; however, this would be accompanied by a decline in enantioselectivity, as the second enantiomer would begin to transform. For this reason, we chose to limit the monitoring of the reaction to 24 hours. The transfer of the optimized process from the analytical to the preparative scale is planned as a continuation of this research and will be the subject of a separate publication in a journal dedicated to organic synthesis. In the present manuscript, under the project's scope, we focus on the optimization of the immobilization procedure and the evaluation of enzymatic activity and stability under analytical-scale conditions.

All together, the results of this paper are worth to be published in IJMS, but only after major revision.

Suggestions: 

  1. A preparative-scale kinetic resolution of racemic 1-phenylethanol through acetylation with isopropenyl acetate in 1,2-dichloropropane (reaction medium) in the presence of CALB immobilized on an octyl-agarose support, under the optimal reaction conditions should be performed and described. Adequate characterization of the isolated products is necessary (exact mass, yield, ee, optical rotation, 1H- and 13C-NMR, HRMS or EA, etc). When a product known in literature, the authors should give the literature data comparatively in the Experimental.

Response to Reviewer Comments: Thank you very much for your valuable suggestions regarding the preparative-scale work. The primary objective of this study is to optimize the immobilization of Candida antarctica lipase B (CALB) on an octyl-agarose support by selecting appropriate conditions, including buffer type (pH and concentration), reaction time, and temperature. Subsequently, the enantioselective activity of the immobilized enzyme, as well as its thermal and storage stability, were evaluated on an analytical scale. As a model reaction, the transesterification of racemic 1-phenylethanol with isopropenyl acetate in 1,2-dichloropropane was employed. This reaction system was used to assess the enzyme’s enantioselectivity and resistance to elevated temperatures, light exposure, and prolonged storage. The main aim of this work was to determine optimal immobilization conditions and to characterize the biocatalyst's performance under controlled analytical-scale conditions. Although the transfer of the developed methodology to a preparative scale is indeed of great importance, such work entails additional technological and experimental challenges. Therefore, in the design of this research project, we deliberately decided to reserve the preparative-scale experiments for a subsequent study. These will be addressed and described in detail in a separate publication, focusing on the isolation and characterization of the reaction products within the context of organic synthesis. To ensure clarity and consistency of the results presented in the current manuscript, we have limited our investigation to analytical-scale experiments. The products and substrates obtained from the biocatalytic reactions were analyzed using HPLC and compared with analytical standards. Additionally, chemical acetylation (with acetyl chloride) of the racemic substrate and its enantiomers was performed. Compound identification was based on a comparison of retention times under identical analytical conditions, including those of commercially available standards and acetylation products.

  1. HPLC chromatograms with baseline separation of the enantiomeric peaks for racemic alcohol and ester should be added (Supporting information). Add also the chromatograms for the enantiomeric products, from the enzymatic reaction (Supporting information).

Response to Reviewer Comments: Thank you for your valuable suggestions. As recommended by the Reviewer, the HPLC chromatogram has been included in the revised manuscript. The chromatogram presents four peaks in a single chromatographic analysis: two corresponding to the substrates and two to the products of the transesterification reaction of racemic 1-phenylethanol with isopropenyl acetate, catalyzed by immobilized Candida antarctica lipase B (CALB).

  1. In the main text (line 338) ”The most optimal chromatographic conditions…“ should be corrected into ”The optimal chromatographic conditions…“; in Tables 1-3, columns Time of kinetic resolution should be deleted and 24 h reaction time included in the Titles). Typos, spelling errors should be corrected too.

Response to Reviewer Comments: Thank you for your valuable comments. The Reviewer’s suggestions have been carefully addressed. The phrase “the most...” has been removed, and the columns containing information on the kinetic resolution time have been eliminated; the relevant details have instead been incorporated into the table caption. In addition, minor typographical and spelling errors have been corrected throughout the manuscript.

The changes made are highlighted in green within the text.

Reviewer 2 Report

Comments and Suggestions for Authors

This work reports CALB-octyl-agarose support as a useful tool in pharmaceutical

biocatalysis by transesterification in organic medium. There are several comments below:

In the literature, there are already several reports using octyl- agarose to immobilize lipase. The authors should give a proper credit to them. Especially, please account for the novelties and new findings of this work.

As discussed in Line 168, what are the enzyme loadings for different conditions in Table 1?

Simple two-point storage testes seem to be not sufficient. How many cycles can the biocatalyst be reused?

Author Response

Authors’ Response to Reviewer 2

Comments and Suggestions for Authors

This work reports CALB-octyl-agarose support as a useful tool in pharmaceutical biocatalysis by transesterification in organic medium. There are several comments below:

  • In the literature, there are already several reports using octyl- agarose to immobilize lipase. The authors should give a proper credit to them. Especially, please account for the novelties and new findings of this work.

Response to Reviewer Comments: Thank you for your valuable suggestions. Under the Reviewer’s recommendation, we have collected and incorporated relevant literature data concerning the use of octyl-agarose for lipase immobilization. A corresponding comment highlighting the novelty and distinct contribution of our study has been added to the manuscript.

The changes made are highlighted in green within the text.

In the manuscript:  line 294 and 270

 We also reviewed recent publications on lipase immobilization on the agarose support (Table 3), identifying the lipases used, supports (with their modifications), substrates for activity assessment, reaction medium, and type of reaction. It should be noted that the literature emphasizes the fact that the effect of agarose beads on lipase activity in organic solvents is poorly described [38]. As can be seen from the prepared table, most studies on the catalytic activity of immobilized lipases (on the discussed support - agarose), particularly CALB, have been performed in an aqueous medium (mainly hydrolysis reactions in buffer). In contrast, our study proposes the use of octyl-agarose-immobilized CALB for reactions in organic solvents. This approach opens new possibilities for conducting biocatalysis by esterification or transesterification, using CALB immobilized on octyl-agarose, offering excellent enantioselectivity. Our protocol for the preparation of octyl-agarose support with CALB for biocatalysis in an organic reaction medium, including immobilization of CALB on octyl-agarose in acidic pH buffer - citrate buffer (pH 4; 300 mM), followed by air-drying, according to our knowledge, has not been widely described in the literature. The immobilized CALB obtained using this method can be introduced into organic media and used for kinetic resolution of (R,S)-1-phenylethanol via transesterification (in DCP; logP value 1.98), with high enantioselectivity. At the same time, we emphasize the potential to modulate the CALB activity by adjusting the pH of the immobilization buffer. Therefore, immobilization conditions (e.g., including buffer pH and concentration) should be selected in relation to the substrate, type of reaction, and organic solvent used in biocatalysis.”

Table 3. List of studies on lipase immobilized on agarose support and application in biocatalysis

Author

Lipase

Support

Substrate

Reaction medium/
type of reaction

Literature

de Andrades et al.

CALA1, CALB

octyl-agarose; amino-hexyl-agarose; amino-octyl-agarose;

p-nitrophenyl ester, triacetin, (R) and (S)-methyl mandelate

aqueous buffer medium/ hydrolysis

[39]

Abdelkader et al.

PLC-BC 2

octyl-sepharose,
Q-sepharose, glyoxyl agarose

pNPPC3

aqueous buffer medium/ hydrolysis

[40]

do Nascimento et al.

FL4

octyl glyoxyl agarose, octyl
monoaminoethyl-N-aminoethyl glutaraldehyde agarose, octyl divinyl sulfone agarose

 pNPB5

aqueous buffer medium/ hydrolysis

[41]

Sabi et al.

ETL6

octyl-agarose

triacetin, pNPB5

aqueous buffer medium/ hydrolysis

[42]

Abellanas-Perez et al.

CALA1, CALB

amino-octyl-vinyl sulfone agarose, amino-hexyl-vinyl sulfone agarose, octyl-vinyl sulfone agarose

triacetin, pNPB5,
(D) and (L)-methyl mandelate

aqueous buffer medium/ hydrolysis

[43]

Arana-Pena et al.

CALA1, CALB, CRL7, RML8

octyl-agarose

triacetin, pNPB5,
(R) and (S)-methyl mandelate

aqueous buffer medium/ hydrolysis

[44]

Arana-Pena et al.

CALB, CALA, TLL9, RML8, LEU10

octyl-agarose, octyl-vinyl sulfone agarose, octyl-vinyl sulfone agarose coated with PEI11

triacetin, pNPB5,
(R) and (S)-methyl mandelate

aqueous buffer medium/ hydrolysis

[45]

Ahrari et al.

 ROL12

octyl-sepharose, cyanogen bromide activated sepharose, glyoxyl-agarose, Q-sepharose, monoaminoethyl-N-aminoethyl-agarose

pNPB5, fish oil

aqueous buffer medium, biphasic system/ hydrolysis

[46]

Siódmiak
et al.

CALB,
CRL-OF14

octyl-agarose

(R,S)-flurbiprofen with methanol

organic solvents/
esterification

[32]

1Lipase A from Candida antarctica, 2Phospholipase C from Bacillus ceresus, 3p-Nitrophenylphosphorylcholin, 4Flaxseed lipase, 5 p-nitrophenylbutyrate 6 Eversa Transform, 7Candida rugosa lipase, 8Rhizomucor miehei lipase, 9Thermomyces lanuginosus lipase, 10Lecitase ultra, 11Polyethylenimine, 12Rhizopus oryzae lipase, 13Lipase OF from Candida rugosa

  • As discussed in Line 168, what are the enzyme loadings for different conditions in Table 1?

Response to Reviewer Comments: Thank you for your valuable suggestions. The enzyme loading values were previously reported in an earlier publication. For clarity and completeness, we have now included these data in Table 1 of the revised manuscript, accompanied by the appropriate citation.

 The changes made are highlighted in green within the text (line 150)

  • Simple two-point storage testes seem to be not sufficient. How many cycles can the biocatalyst be reused?

Response to Reviewer Comments: Thank you for your valuable suggestions. We have entered additional data into the manuscript.

The changes made are highlighted in green within the text (line 308, 374, 395)

“CALB – Stability Study

2.2.1. Thermal stability, Storage stability and Operational stability study of CALB”

“To evaluate the thermal stability of the catalytic protein, a thermal stability study of the CALB in its dry form was performed in a climatic chamber. The studies were carried out for 7 days. In the climatic chamber, the influence of a temperature of 65°C was determined, and additionally, under the study, the effect of visible light (wavelength range: 400-800nm) was also examined. For comparison with extremely high temperatures, a parallel experiment was performed at 4°C in a refrigerator, without exposure to visible light. This setting was used as a control to assess the impact of high temperature (65°C) and light (400-800nm) on the thermal stability of the immobilized enzyme.”

“In the next step, a storage stability study of CALB was conducted. The immobilized CALB, using the same immobilization procedure as in the thermal stability study, was stored in dry form in two conditions: at 4°C in a refrigerator (without light in the visible spectral range) and at room temperature (22°C, without light in the visible spectral range) for 28 days. Storage stability was monitored at four time points: after 7, 14, 21, and 28 days of storage. The assessment of storage stability was carried out using the same reaction conditions as in the thermal stability study. The results of the studies (Figure 3) indicate good stability of the catalytic system under both storage conditions throughout the 28-day period, stored both at 4°C and 22°C.”

“As part of the project, operational stability studies were also conducted by reusing the biocatalyst in subsequent catalytic cycles of the transesterification of (R,S)-1-phenylethanol. After the third cycle, the conversion of the reaction was C = 10±0.3%, which indicates a significant decrease in conversions compared to the initial value (C = 43.1±0.5% from the first cycle). We believe that the obtained results were influenced by technological difficulties with support recovery from reaction medium, the analytical scale of the experiments (50 mg of support), and the nature of the interactions between CALB and the support (suggesting possible enzyme leakage). Immobilization of lipase was performed in a citrate buffer (pH 4; 300 mM).”

Reviewer 3 Report

Comments and Suggestions for Authors

The manuscript reports the immobilization of lipase B from Candida antarctica (the most frequently used microbial lipase) on octyl-agarose support (Octyl-Sepharose CL-4B, commercially available), the optimization of the immobilization conditions and evaluation of the storage stability of this biocatalyst in different temperature and light conditions. The model reaction selected for this study was the kinetic resolution of (R,S)-1-phenylethanol by enantioselective transesterification with isopropenyl acetate, using 1,2-dichloropropane as reaction medium. Although the topic of enzyme stabilization by immobilization is of scientific interest, I cannot recommend this manuscript for publication in the present form.

  1. The reported research lacks novelty. The enzyme, support, and immobilization method are very common and were explored by the authors of this manuscript, as well (references #30-32). Therefore, this work should provide consistent improvements compared to previous reports, but in my opinion it is not the case. The optimization of the immobilization conditions at three different pH values (the buffer concentration has no effect) is not sufficient to recommend the manuscript for publication in a Q1-ranked journal. Concerning the storage stability, its evaluation for only 7 days is totally non-relevant, since immobilized lipases are usually stable for month and even years when stored in a refrigerator (we noticed that in our laboratory several times). Operational stability across many reuse cycles would have been much more relevant in this respect, because enzyme leakage can be a real problem (it was not mentioned, neither studied).
  2. The authors should provide a table with the main literature results concerning the immobilization of CALB lipase on octyl-agarose supports, compared with the results of this work, to demonstrate that the present results show a consistent improvement.
  3. The selection of 1,2-dichloropropane as reaction medium is not explained in the manuscript. Chlorinated organic solvents are harmful for the environment and toxic. Particularly, 1,2-dicholopropane is classified as a possible carcinogen https://www.cdc.gov/niosh/idlh/78875.html. Many other less toxic organic solvents provided excellent results in the kinetic resolution of (R,S)-1-phenylethanol, like as n-hexane.
  4. The extremely low conversion obtained in the reaction catalyzed by the free CALB lipase (about 1% in 24 h) is doubtful for me. The authors did not provide the immobilization yield (assessed as loaded protein) but using 10 mg free and 50 mg immobilized lipase, respectively, cannot result in a 40-fold higher conversion for the immobilized form. I do not remember such an activation reported following the immobilization and it is very unlikely because it is very difficult to explain. Probably, something happened in the reaction system with the free lipase, inhibiting the enzyme or impeding the reaction. We frequently tested free CALB lipases in the kinetic resolution of (R,S)-1-phenylethanol obtaining excellent conversions and >99% ee. Can the authors give a link to this lipase on the website of the producer (ChiralVision), to get the specification of the product?

Author Response

Authors’ Response to Reviewer 3

Comments and Suggestions for Authors

The manuscript reports the immobilization of lipase B from Candida antarctica (the most frequently used microbial lipase) on octyl-agarose support (Octyl-Sepharose CL-4B, commercially available), the optimization of the immobilization conditions and evaluation of the storage stability of this biocatalyst in different temperature and light conditions. The model reaction selected for this study was the kinetic resolution of (R,S)-1-phenylethanol by enantioselective transesterification with isopropenyl acetate, using 1,2-dichloropropane as reaction medium. Although the topic of enzyme stabilization by immobilization is of scientific interest, I cannot recommend this manuscript for publication in the present form.

  1. The reported research lacks novelty. The enzyme, support, and immobilization method are very common and were explored by the authors of this manuscript, as well (references #30-32). Therefore, this work should provide consistent improvements compared to previous reports, but in my opinion it is not the case. The optimization of the immobilization conditions at three different pH values (the buffer concentration has no effect) is not sufficient to recommend the manuscript for publication in a Q1-ranked journal. Concerning the storage stability, its evaluation for only 7 days is totally non-relevant, since immobilized lipases are usually stable for month and even years when stored in a refrigerator (we noticed that in our laboratory several times). Operational stability across many reuse cycles would have been much more relevant in this respect, because enzyme leakage can be a real problem (it was not mentioned, neither studied).

  Response to Reviewer Comments: Thank you for your valuable suggestions. We have collected and incorporated relevant literature data concerning the use of octyl-agarose for lipase immobilization. A corresponding comment highlighting the novelty and distinct contribution of our study has been added to the manuscript. Under the Reviewer’s recommendation, we presented a comprehensive scope of CALB optimization studies on octyl-agarose, and thermal and storage stability assessments, as well as evaluations of operational stability. In this study, we presented the results of the optimization of buffer pH and concentration used during the immobilization process. We demonstrated that there is no universal pH value or buffer concentration that is optimal for all applications in organic solvents. For example, in the case of racemic 1-phenylethanol, an acidic buffer pH proved to be optimal, whereas for racemic flurbiprofen (as reported in our previous paper), an alkaline buffer was more effective. Moreover, while buffer concentration had a negligible impact on the immobilization of CALB for 1-phenylethanol, it significantly influenced the outcome for flurbiprofen. These findings underscore the necessity for continuous optimization of immobilization conditions tailored to the specific reaction and substrates involved, as discussed in the manuscript. Notably, the literature offers limited information on the optimization of CALB immobilization on octyl-agarose for subsequent application in organic solvents, highlighting the novelty of our research. As part of this optimization, we also examined the effects of immobilization time and temperature, which are detailed in the manuscript. The improvement in immobilization lies in demonstrating—using flurbiprofen and 1-phenylethanol as examples—that both the pH value and concentration of the immobilization buffer play a critical role in achieving high enantioselectivity in organic solvents. Had we applied the protocol optimized for flurbiprofen (alkaline pH) to 1-phenylethanol, the conversion would have been approximately two times lower. This indicates a significant advancement over previously established conditions and underscores the importance of tailoring immobilization parameters to specific substrates. While the majority of studies on the catalytic performance of immobilized lipases—particularly CALB immobilized on agarose supports—have focused on aqueous systems (primarily hydrolysis reactions in buffered media), our work presents a different approach. Specifically, we propose the application of octyl-agarose-immobilized CALB in organic solvents. This strategy broadens the potential of biocatalysis, enabling efficient esterification and transesterification reactions with high enantioselectivity, thereby extending the applicability of immobilized (on agarose) enzymes in organic solvents.

The changes made are highlighted in green within the text. In the manuscript:  line 294 and 270

Table 3. List of studies on lipase immobilized on agarose support and application in biocatalysis

 We also reviewed recent publications on lipase immobilization on the agarose support (Table 3), identifying the lipases used, supports (with their modifications), substrates for activity assessment, reaction medium, and type of reaction. It should be noted that the literature emphasizes the fact that the effect of agarose beads on lipase activity in organic solvents is poorly described [38]. As can be seen from the prepared table, most studies on the catalytic activity of immobilized lipases (on the discussed support - agarose), particularly CALB, have been performed in an aqueous medium (mainly hydrolysis reactions in buffer). In contrast, our study proposes the use of octyl-agarose-immobilized CALB for reactions in organic solvents. This approach opens new possibilities for conducting biocatalysis by esterification or transesterification, using CALB immobilized on octyl-agarose, offering excellent enantioselectivity. Our protocol for the preparation of octyl-agarose support with CALB for biocatalysis in an organic reaction medium, including immobilization of CALB on octyl-agarose in acidic pH buffer - citrate buffer (pH 4; 300 mM), followed by air-drying, according to our knowledge, has not been widely described in the literature. The immobilized CALB obtained using this method can be introduced into organic media and used for kinetic resolution of (R,S)-1-phenylethanol via transesterification (in DCP; logP value 1.98), with high enantioselectivity. At the same time, we emphasize the potential to modulate the CALB activity by adjusting the pH of the immobilization buffer. Therefore, immobilization conditions (e.g., including buffer pH and concentration) should be selected in relation to the substrate, type of reaction, and organic solvent used in biocatalysis.”

The changes made are highlighted in green within the text. In the manuscript:  Line 255

“As part of the preliminary studies in our laboratory, the immobilization time was also optimized. Samples in a citrate buffer (pH 4, 300 mM) were mixed for 5 min at room temperature and then kept at 4 °C for 2, 8, 10, 14, and 20 hours. The results of the conducted model reaction (transesterification of (R,S)-1-phenylethanol) indicated that the optimal immobilization time was 14 h, as this condition ensured the highest conversion values. Extending the incubation time beyond 14 hours did not lead to further increases in conversion and enantioselectivity (after 2 h of immobilization – C = 14.3±0.2%; after 8h – C = 30.3±1.1%; after 10h – C = 38.4±1.3%, after 14 h – C = 43.1±0.5 %, after 20h – C = 43.6±1.9%). Another parameter optimized was the immobilization temperature. Samples (in citrate buffer at pH 4, 300 mM) were mixed for 5 min at room temperature, followed by incubation (CALB) for 14 hours at 4 °C, 22 °C, and 37 °C. The analyses demonstrated that the highest conversion values of kinetic resolution were achieved when immobilization was performed in a refrigerator (4°C). Conversion values at the other temperatures were slightly lower than those gained from lipase immobilized in a refrigerator (C = 41.1±1.0% in 22 °C and C = 40.4±0.8% in 37 °C).

The changes made are highlighted in green within the text. In the manuscript:  Line 144

During the preliminary phase of the study, we investigated the effect of various organic solvents as a reaction medium (n-hexane, n-heptane, dichloromethane (DCM), 1,2-dichloroethane (DCE), 1,2-dichloropropane (DCP), diisopropyl ether (DIPE), and t-butyl methyl ether (MTBE) on the assessed catalytic parameters of immobilized CALB (data not shown). Among the tested solvents, the best results were obtained for DCP; hence, this solvent was used in further studies.

The changes made are highlighted in green within the text (line 308, 374, 395)

“CALB – Stability Study

2.2.1. Thermal stability, Storage stability and Operational stability study of CALB”

“To evaluate the thermal stability of the catalytic protein, a thermal stability study of the CALB in its dry form was performed in a climatic chamber. The studies were carried out for 7 days. In the climatic chamber, the influence of a temperature of 65°C was determined, and additionally, under the study, the effect of visible light (wavelength range: 400-800nm) was also examined. For comparison with extremely high temperatures, a parallel experiment was performed at 4°C in a refrigerator, without exposure to visible light. This setting was used as a control to assess the impact of high temperature (65°C) and light (400-800nm) on the thermal stability of the immobilized enzyme.”

“In the next step, a storage stability study of CALB was conducted. The immobilized CALB, using the same immobilization procedure as in the thermal stability study, was stored in dry form in two conditions: at 4°C in a refrigerator (without light in the visible spectral range) and at room temperature (22°C, without light in the visible spectral range) for 28 days. Storage stability was monitored at four time points: after 7, 14, 21, and 28 days of storage. The assessment of storage stability was carried out using the same reaction conditions as in the thermal stability study. The results of the studies (Figure 3) indicate good stability of the catalytic system under both storage conditions throughout the 28-day period, stored both at 4°C and 22°C.”

“As part of the project, operational stability studies were also conducted by reusing the biocatalyst in subsequent catalytic cycles of the transesterification of (R,S)-1-phenylethanol. After the third cycle, the conversion of the reaction was C = 10±0.3%, which indicates a significant decrease in conversions compared to the initial value (C = 43.1±0.5% from the first cycle). We believe that the obtained results were influenced by technological difficulties with support recovery from reaction medium, the analytical scale of the experiments (50 mg of support), and the nature of the interactions between CALB and the support (suggesting possible enzyme leakage). Immobilization of lipase was performed in a citrate buffer (pH 4; 300 mM).”

  1. The authors should provide a table with the main literature results concerning the immobilization of CALB lipase on octyl-agarose supports, compared with the results of this work, to demonstrate that the present results show a consistent improvement.

Response to Reviewer Comments: Thank you for your valuable suggestions. Under the Reviewer’s recommendation, we have collected and incorporated relevant literature data concerning the use of octyl-agarose for lipase immobilization. A corresponding comment highlighting the novelty and distinct contribution of our study has been added to the manuscript.

The changes made are highlighted in green within the text.

In the manuscript:  line 294 and 270

Table 3. List of studies on lipase immobilized on agarose support and application in biocatalysis

Author

Lipase

Support

Substrate

Reaction medium/
type of reaction

Literature

de Andrades et al.

CALA1, CALB

octyl-agarose; amino-hexyl-agarose; amino-octyl-agarose;

p-nitrophenyl ester, triacetin, (R) and (S)-methyl mandelate

aqueous buffer medium/ hydrolysis

[39]

Abdelkader et al.

PLC-BC 2

octyl-sepharose,
Q-sepharose, glyoxyl agarose

pNPPC3

aqueous buffer medium/ hydrolysis

[40]

do Nascimento et al.

FL4

octyl glyoxyl agarose, octyl
monoaminoethyl-N-aminoethyl glutaraldehyde agarose, octyl divinyl sulfone agarose

 pNPB5

aqueous buffer medium/ hydrolysis

[41]

Sabi et al.

ETL6

octyl-agarose

triacetin, pNPB5

aqueous buffer medium/ hydrolysis

[42]

Abellanas-Perez et al.

CALA1, CALB

amino-octyl-vinyl sulfone agarose, amino-hexyl-vinyl sulfone agarose, octyl-vinyl sulfone agarose

triacetin, pNPB5,
(D) and (L)-methyl mandelate

aqueous buffer medium/ hydrolysis

[43]

Arana-Pena et al.

CALA1, CALB, CRL7, RML8

octyl-agarose

triacetin, pNPB5,
(R) and (S)-methyl mandelate

aqueous buffer medium/ hydrolysis

[44]

Arana-Pena et al.

CALB, CALA, TLL9, RML8, LEU10

octyl-agarose, octyl-vinyl sulfone agarose, octyl-vinyl sulfone agarose coated with PEI11

triacetin, pNPB5,
(R) and (S)-methyl mandelate

aqueous buffer medium/ hydrolysis

[45]

Ahrari et al.

 ROL12

octyl-sepharose, cyanogen bromide activated sepharose, glyoxyl-agarose, Q-sepharose, monoaminoethyl-N-aminoethyl-agarose

pNPB5, fish oil

aqueous buffer medium, biphasic system/ hydrolysis

[46]

Siódmiak
et al.

CALB,
CRL-OF14

octyl-agarose

(R,S)-flurbiprofen with methanol

organic solvents/
esterification

[32]

1Lipase A from Candida antarctica, 2Phospholipase C from Bacillus ceresus, 3p-Nitrophenylphosphorylcholin, 4Flaxseed lipase, 5 p-nitrophenylbutyrate 6 Eversa Transform, 7Candida rugosa lipase, 8Rhizomucor miehei lipase, 9Thermomyces lanuginosus lipase, 10Lecitase ultra, 11Polyethylenimine, 12Rhizopus oryzae lipase, 13Lipase OF from Candida rugosa

We also reviewed recent publications on lipase immobilization on the agarose support (Table 3), identifying the lipases used, supports (with their modifications), substrates for activity assessment, reaction medium, and type of reaction. It should be noted that the literature emphasizes the fact that the effect of agarose beads on lipase activity in organic solvents is poorly described [38]. As can be seen from the prepared table, most studies on the catalytic activity of immobilized lipases (on the discussed support - agarose), particularly CALB, have been performed in an aqueous medium (mainly hydrolysis reactions in buffer). In contrast, our study proposes the use of octyl-agarose-immobilized CALB for reactions in organic solvents. This approach opens new possibilities for conducting biocatalysis by esterification or transesterification, using CALB immobilized on octyl-agarose, offering excellent enantioselectivity. Our protocol for the preparation of octyl-agarose support with CALB for biocatalysis in an organic reaction medium, including immobilization of CALB on octyl-agarose in acidic pH buffer - citrate buffer (pH 4; 300 mM), followed by air-drying, according to our knowledge, has not been widely described in the literature. The immobilized CALB obtained using this method can be introduced into organic media and used for kinetic resolution of (R,S)-1-phenylethanol via transesterification (in DCP; logP value 1.98), with high enantioselectivity. At the same time, we emphasize the potential to modulate the CALB activity by adjusting the pH of the immobilization buffer. Therefore, immobilization conditions (e.g., including buffer pH and concentration) should be selected in relation to the substrate, type of reaction, and organic solvent used in biocatalysis.”

  1. The selection of 1,2-dichloropropane as reaction medium is not explained in the manuscript. Chlorinated organic solvents are harmful for the environment and toxic. Particularly, 1,2-dicholopropane is classified as a possible carcinogen https://www.cdc.gov/niosh/idlh/78875.html. Many other less toxic organic solvents provided excellent results in the kinetic resolution of (R,S)-1-phenylethanol, like as n-hexane.

Response to Reviewer Comments: Thank you for your valuable suggestions. The selection of the solvent was based on the enantioselective activity values of the immobilized lipase observed during preliminary studies. Regarding the potential toxicity of DCP, we fully agree with the reviewer’s concern and intend to optimize future reactions using solvents with lower toxicity. We would also like to emphasize that the reactions were carried out on an analytical scale, with the solvent volume in each reaction limited to only 410 μL.

The changes made are highlighted in green within the text.

In the manuscript:  line 144

„During the preliminary phase of the study, we investigated the effect of various organic solvents as a reaction medium (n-hexane, n-heptane, dichloromethane (DCM), 1,2-dichloroethane (DCE), 1,2-dichloropropane (DCP), diisopropyl ether (DIPE), and t-butyl methyl ether (MTBE) on the assessed catalytic parameters of immobilized CALB (data not shown). Among the tested solvents, the best results were obtained for DCP; hence, this solvent was used in further studies.”

  1. The extremely low conversion obtained in the reaction catalyzed by the free CALB lipase (about 1% in 24 h) is doubtful for me. The authors did not provide the immobilization yield (assessed as loaded protein) but using 10 mg free and 50 mg immobilized lipase, respectively, cannot result in a 40-fold higher conversion for the immobilized form. I do not remember such an activation reported following the immobilization and it is very unlikely because it is very difficult to explain. Probably, something happened in the reaction system with the free lipase, inhibiting the enzyme or impeding the reaction. We frequently tested free CALB lipases in the kinetic resolution of (R,S)-1-phenylethanol obtaining excellent conversions and >99% ee. Can the authors give a link to this lipase on the website of the producer (ChiralVision), to get the specification of the product?

Response to Reviewer Comments: We sincerely thank the Reviewer for the insightful comments regarding the use of free lipase in the transesterification reaction of (R,S)-1-phenylethanol. In light of these suggestions, we have decided to focus exclusively on the description of lipase activity in its immobilized form in the revised manuscript, and we have made the corresponding modifications accordingly. We will further investigate the performance of the free enzyme to determine whether any inhibitory or limiting factors, as suggested by the Reviewer, may have influenced the results. These findings will be thoroughly addressed in a forthcoming study dedicated to the behavior of CALB in its free form and described in another paper. With regard to the reaction yield (immobilization yield assessed as loaded protein), we have included the relevant information in the revised manuscript.

We are including a photo of the bottle with CALB.

The changes made are highlighted in green within the text. In the manuscript:  line 166

“The immobilization yield (Iy) was calculated: in the case with the use procedure immobilization with citrate buffer (pH 4, 100 mM) immobilization yield was Iy = 32.4±0.4%, using phosphate buffer (pH 7, 100 mM) Iy = 21.1±0.7% and with tris-base buffer (pH 9, 100 mM) Iy = 15.3±0.3%. “

Round 2

Reviewer 1 Report

Comments and Suggestions for Authors

The authors have fully implemented the changes, additions, and corrections suggested by the Reviewer. In view of the preparative-scale KR, the authors wrote: “Although the transfer of the developed methodology to a preparative scale is indeed of great importance, such work entails additional technological and experimental challenges. Therefore, in the design of this research project, we deliberately decided to reserve the preparative-scale experiments for a subsequent study. These will be addressed and described in detail in a separate publication, focusing on the isolation and characterization of the reaction products within the context of organic synthesis. To ensure clarity and consistency of the results presented in the current manuscript, we have limited our investigation to analytical-scale experiments.”

I suggest acceptance of this Article for publication in IJMS, in the form as it is.

Author Response

Authors’ Response to Reviewer 1

 Comments and Suggestions for Authors:

The authors have fully implemented the changes, additions, and corrections suggested by the Reviewer. In view of the preparative-scale KR, the authors wrote: “Although the transfer of the developed methodology to a preparative scale is indeed of great importance, such work entails additional technological and experimental challenges. Therefore, in the design of this research project, we deliberately decided to reserve the preparative-scale experiments for a subsequent study. These will be addressed and described in detail in a separate publication, focusing on the isolation and characterization of the reaction products within the context of organic synthesis. To ensure clarity and consistency of the results presented in the current manuscript, we have limited our investigation to analytical-scale experiments.”

I suggest acceptance of this Article for publication in IJMS, in the form as it is.

 Response to Reviewer Comments:

We would like to express our sincere gratitude for your thorough review of our manuscript. We deeply appreciate the time and effort you dedicated to evaluating our work, as well as your insightful and constructive comments. Your suggestions have been extremely helpful in enhancing the clarity and overall quality of the manuscript. We are also truly grateful for your recommendation to accept our manuscript for publication.

Reviewer 2 Report

Comments and Suggestions for Authors

The manuscript can be accepted for publication.

Author Response

Authors’ Response to Reviewer 2

 Comments and Suggestions for Authors:

The manuscript can be accepted for publication.

Response to Reviewer Comments:

We sincerely thank you for your careful and thoughtful review of our manuscript. We truly appreciate the time and effort you devoted to evaluating our paper, as well as the constructive and insightful feedback you provided. Your comments have been instrumental in improving the clarity and overall quality of the manuscript. We are also very grateful for your recommendation to accept our manuscript for publication.

Reviewer 3 Report

Comments and Suggestions for Authors

I appreciate that the manuscript was consistently revised and improved compared to the original submission. The added value of this contribution is more clearly highlighted, and the comparative evaluation with previous studies was also accomplished. I can accept the answers of the authors to my observations and questions, although the selection of 1,2-dichloropropane as reaction medium is still questionable in my opinion. Consequently, my recommendation is to accept the revised manuscript for publication, with the following minor recommendations:

  1. I noticed that the title was modified, but the statement “pharmaceutical transesterification” is still nor appropriate, because there is a single transesterification reaction in chemistry. I recommend changing the title as “CALB Immobilized on Octyl-Agarose - an Efficient Pharmaceutical Biocatalyst for Transesterification in Organic Medium”.
  2. Please add in Table 3 (at the end) the results of the present study, allowing an easier comparison with previous reports.
  3. I also recommend changing the legend of Table 3 to “Comparative evaluation of the studies directing the immobilization of lipases on agarose supports” (or a similar formulation).

Author Response

Authors’ Response to Reviewer 3

Comments and Suggestions for Authors

I appreciate that the manuscript was consistently revised and improved compared to the original submission. The added value of this contribution is more clearly highlighted, and the comparative evaluation with previous studies was also accomplished. I can accept the answers of the authors to my observations and questions, although the selection of 1,2-dichloropropane as reaction medium is still questionable in my opinion. Consequently, my recommendation is to accept the revised manuscript for publication, with the following minor recommendations:

  1. I noticed that the title was modified, but the statement “pharmaceutical transesterification” is still nor appropriate, because there is a single transesterification reaction in chemistry. I recommend changing the title as “CALB Immobilized on Octyl-Agarose - an Efficient Pharmaceutical Biocatalyst for Transesterification in Organic Medium”.
  2. Please add in Table 3 (at the end) the results of the present study, allowing an easier comparison with previous reports.
  3. I also recommend changing the legend of Table 3 to “Comparative evaluation of the studies directing the immobilization of lipases on agarose supports” (or a similar formulation).

Response to Reviewer Comments:

We sincerely appreciate your thorough and insightful review of our manuscript. Thank you for dedicating your time to evaluating our work and for the valuable comments and suggestions you provided. Your feedback has significantly contributed to improving the clarity and overall quality of the manuscript. We are also very grateful for your recommendation to accept our manuscript for publication. Your suggestions have been carefully addressed and incorporated into the revised version.

1. I noticed that the title was modified, but the statement “pharmaceutical transesterification” is still nor appropriate, because there is a single transesterification reaction in chemistry. I recommend changing the title as “CALB Immobilized on Octyl-Agarose - an Efficient Pharmaceutical Biocatalyst for Transesterification in Organic Medium”.

 Response to Reviewer Comments: Thank you for your valuable suggestion. In accordance with the reviewer’s suggestion, the title of the manuscript has been revised as follows:

“CALB Immobilized on Octyl-Agarose - an Efficient Pharmaceutical Biocatalyst for Transesterification in Organic Medium”
The changes made are highlighted in blue within the text (line 2).

2. Please add in Table 3 (at the end) the results of the present study, allowing an easier comparison with previous reports.

Response to Reviewer Comments: Thank you for your valuable suggestion. As recommended, the results of the present study have been added at the end of Table 3. The corresponding changes are highlighted in blue in the revised manuscript (line 296).

3. I also recommend changing the legend of Table 3 to “Comparative evaluation of the studies directing the immobilization of lipases on agarose supports” (or a similar formulation).

 Response to Reviewer Comments: Thank you for your valuable suggestion. In response, the legend of Table 3 has been revised to read:
“Comparative evaluation of the studies directing the immobilization of lipases on agarose supports.”
The revision is highlighted in blue in the manuscript (line 294).